# Factors Related to Underweight Prevalence among 33,776 Children Below 60 Months Old Living in Northern Geopolitical Zones, Nigeria (2008–2018)

**DOI:** 10.3390/nu14102042

**Published:** 2022-05-13

**Authors:** Piwuna C. Goson, Tanko Ishaya, Osita K. Ezeh, Gladys H. Oforkansi, David Lim, Kingsley E. Agho

**Affiliations:** 1Department of Psychiatry, College of Health Sciences, University of Jos, Jos 930003, Nigeria; piwunag@unijos.edu.ng; 2Department of Computer Science, University of Jos, Jos 930003, Nigeria; ishayat@unijos.edu.ng; 3School of Health Sciences, Western Sydney University, Locked Bag 1797, Penrith, NSW 2750, Australia; ezehosita@yahoo.com (O.K.E.); david.lim@westernsydney.edu.au (D.L.); 4Department of Economics, Nnamdi Azikiwe University, Awka 420218, Nigeria; okwudilijudith@yahoo.com; 5Translational Health Research Institute (THRI), Campbelltown Campus, Western Sydney University, Penrith, NSW 2571, Australia

**Keywords:** underweight in children, northern geopolitical zone, children below 60 months old, trends in underweight, underweight prevalence, multilevel logistics regression

## Abstract

The prevalence of underweight among children below 60 months old in Nigeria remains a significant public health challenge, especially in northern geopolitical zones (NGZ), ranging from 15% to 35%. This study investigates time-based trends in underweight prevalence and its related characteristics among NGZ children below 60 months old. Extracted NGZ representative dataset of 33,776 live births from the Nigeria Demographic and Health Survey between 2008 and 2018 was used to assess the characteristics related to underweight prevalence in children aged 0–23, 24–59, and 0–59 months using multilevel logistics regression. Findings showed that 11,313 NGZ children below 60 months old were underweight, and 24–59-month-old children recorded the highest prevalence (34.8%; 95% confidence interval: 33.5–36.2). Four factors were consistently significantly related to underweight prevalence in children across the three age groups: poor or average-income households, maternal height, children who had diarrhoea episodes, and children living in the northeast or northwest. Intervention initiatives that include poverty alleviation through cash transfer, timely health checks of offspring of short mothers, and adequate clean water and sanitation infrastructure to reduce the incidence of diarrhoea can substantially reduce underweight prevalence among children in NGZ in Nigeria.

## 1. Introduction

Underweight prevalence—especially in vulnerable groups, such as children aged 0–59 months—can hamper these children’s survival and mental and cognitive development, which often leads to increased morbidity and mortality [1]. This condition is largely attributed to undernutrition. Evidence from a recent United Nations Children’s Fund (UNICEF) indicated that undernutrition accounts for almost 50% of all mortality in children below 60 months old, particularly in Africa and Asia [2]. The adverse impacts of undernutrition in children can be long-lasting, devastating, and often irreversible, such as neurocognitive delay, growth impediments leading to short adult stature, lower productivity later in life, and poor learning performance [3,4]. Victora and colleagues also suggested in their prospective cohort studies from five developing countries (Brazil, Guatemala, India, the Philippines, and South Africa) that undernourished children have higher odds to become short adults, have lower educational achievement, lower economic status in adulthood, and higher probability of given birth to smaller infants [5]. An underweight child is designated as one whose weight-for-age z (WAZ) score is < −2 standard deviations (SD) of the World Health Organisation (WHO) child growth standard [6]. Underweight prevalence is a combined indicator that covers both wasting and stunting [7], indicating that both or each can be mirrored by underweight prevalence [8].

In Nigeria, underweight prevalence among children below 60 months old remains a significant public health challenge; for instance, only an 8.3% decline was recorded in the past decade and a half, from 24% in 2003 to 22% in 2018 [9,10,11]. This 8.3% unassertive improvement may be attributable to a greater percentage of children below 60 months old who had vitamin A supplements and deworming medication [11]. Despite this modest national decrease, underweight prevalence is widespread at the subnational level, particularly in northern geopolitical zones (NGZ), which comprise the northcentral (NC), northeast (NE), and northwest (NW). A recent childhood anthropometric report indicated that 15%, 30%, and 35% of children below 60 months old residing in NC, NE, and NW, respectively, were underweight between 2013 and 2018 as compared to the southern zone children, ranging from 10% to 15% [11]. This high underweight prevalence implies that a remarkable number of children in NGZ, especially the NE and NW, were deprived of rich nutritional foods and high-protein-energy nutrients, which might have led to increased nutritional deficiency among NGZ children, and this subsequently necessitated the conception of this study.

The characteristics related to underweight prevalence have been extensively studied among children below 60 months old, especially in less-developed countries, such as Ethiopia [12], Bangladesh [13], and Indonesia [14]. However, the literature is inadequate in Nigeria. The only studies conducted were small-scale hospital [15] or community-based studies [16,17] except for Akombi et al.’s [18] study, which specifically used aggregated national representative data to investigate characteristics related to wasting and underweight prevalence among children below 60 months old in Nigeria. These studies suggested that characteristics such as diarrhoea episodes, low maternal or paternal education, perceived small child body size at birth, being a male child, home delivery, family size greater than six, non-exclusive breastfeeding, and fever are increasingly associated with underweight prevalence among children younger than five years. The limitations of these studies included a lack of information regarding changes over time and the fact that the aggregated nationwide estimates could cover the inequalities in the demographic, health, social, and economic concerns of the NGZ. Changes over time among underweight children are crucial because they can assist the considered geopolitical zones in promoting the effectiveness of past underweight interventional coverage and, subsequently, guide the strengthening of current or future intervention strategies. As argued previously, using regional or zonal data can produce enhanced estimates for adequate interventional policy design and operation [19] as opposed to aggregated nationwide estimates, which would mask widespread gaps in the NGZ given dissimilarities in culture, religion, demographic, and socioeconomic development within and across Nigeria’s states. Additionally, children born at home or not hospitalised were not included in the hospital-based studies even though most deliveries in Nigeria are conducted at home [11]. This indicates that the generated estimates may be ineffective in designing effective interventional policies across wider geopolitical zones and/or states. Therefore, using disaggregated regional or geopolitical zone-specific data can unmask complex entwined contextual factors that differentially impact interventional initiatives on underweight children across communities and/or states. Combining geopolitical zones with akin features (i.e., socioeconomic, ethnic, cultural, and religious beliefs), such as those of NGZ, can unrestrainedly report differences and unhindered interventional coverage efficacy. No disaggregated population-based studies have examined the odds of the relation of independent characteristics with underweight prevalence in children below 60 months old living in NGZ.

Consequently, this study investigated the likely characteristics related to underweight in children below 60 months old living in NGZ in a mutually exclusive disaggregated age category (0–23 and 24–59 months old) and aggregated cumulative age group of 0–59 months using the extracted NGZ dataset from the Nigeria Demographic and Health Survey (NDHS) dataset for 2008, 2013, and 2018. Additionally, changes over time in the prevalence of underweight children aged 0–23, 24–59, and 0–59 months in the geopolitical zone and its state level were examined. Findings from the standardised disaggregated national representative data would equip government and non-governmental organisations with adequate evidence-based information in formulating appropriate zone-specific, cost-effective intervention initiatives to scale down underweight children in Nigeria.

## 2. Materials and Methods

The NDHS 2008, 2013, and 2018 standardised national representative surveys were combined, and data related to the NGZ were extracted for analysis. Structured questionnaires were used to gather data regarding the health and demographic characteristics of the child and mother, including anthropometry data during the surveys. Women aged between 15 and 49 years who were interviewed during the surveys detailed the live births of their children. In all the surveys considered, approximately 62,169 live births of children less than 60 months old occurred in the NGZ; of these, 17,184 were from the 2008 NDHS, 21,693 were from the 2013 NDHS, and 23,292 were from the 2018 NDHS.

A digital display scale, particularly that of SECA 878U, was used to measure the children’s weight. The weight of children was documented using the standardised weight-for-age measurement procedure as described by the WHO [6]. In the pooled surveys, 33,776 children below 60 months old who had comprehensive and valid data concerning the birth date and weight measurements in the NGZ were used for the study analysis. The statistical methodology used to record live births and the anthropometric guidelines regarding the weight measurement of children below 60 months old have been detailed elsewhere [9,10,11].

### 2.1. Dependent Variable

The dependent variables were underweight in children below 60 months old and disaggregated into three age categories in months, that is, 0–23, 24–59, and 0–59. Underweight cases arising within the study age groups were measured twofold: the case of underweight was estimated as the WAZ score < −2 SD and coded as 1, and non-case underweight with WAZ ≥ −2 SD was coded as 0.

### 2.2. Possible Related Confounding Characteristics

A recent nutritional framework described by the UNICEF [7], and past studies conducted in developing countries [12,13,14] were used to identify the possible confounding factors to be examined in the current study. The dependent variables were investigated against all 29 potential confounding variables selected, which were categorised into seven distinct classes (Table 1).

The UNICEF framework entails direct immediate characteristics, which include child nutrition and disease occurrence. It has been previously suggested that poor child nutrition and recurrent child illness increase the association with underweight nutrition. Dietary diversity score (DDS) mirrors the prevalence of eight possible food categories taken by a child in the last 24 h in children younger than five years [15,16,17]. Feeding practices and DDS were parameters used to measure adequate child nutrition prior to the survey interview, and the food categories were classified into two classes in the study analysis (child consumed ≥ five food categories and child consumed < five food categories). These eight food groups have been reported elsewhere [11]. Recurrent child illnesses (e.g., diarrhoea and fever) were possible disease occurrences in the last 14 days before the survey interview date.

According to the earlier literature [12,14,18,20], socioeconomic characteristics (e.g., educational attainment by mothers/fathers, economic status of a household, mother’s work status, and the number of wives or women living in a household) are associated with underweight prevalence in children below 60 months old. Household income or expenditure data were not used to quantify the economic status of households due to unavailability of data; however, a self-reported household asset-based factor score was utilised as a wealth proxy measure to categorise the household economic status using principal component analysis [21]. This means weights were assigned to the self-identified assets, and these assets have been listed in the NDHS report [11]. In the three combined surveys, the household wealth index factor scores were classified into three groups: poor, middle, and rich households.

Individual child and maternal characteristics are increasingly associated with underweight prevalence in children below 60 months old [12,18]. The child and maternal characteristics incorporated are presented in Table 1. Maternal autonomies [22], such as having healthcare, earning/financial, and movement autonomies, were considered, and these autonomies were grouped as household decision-related characteristics. Additionally, included in the study analysis was maternal access to electronic or print media classified as knowledge of healthcare through media. Healthcare service-related characteristics (e.g., mode of birth, birth assistance, and birthplace) and community-level characteristics (type of residence and geopolitical zone) were also included in this study. Table 1 depicts the classification of all the independent characteristics used in the study analysis.

### 2.3. Data Analysis

For each wave of the surveys, the frequency distribution and underweight prevalence of children below 60 months old for all potential associated independent variables listed in Appendix A were estimated with a confidence interval (CI) of 95%. Data from all surveys considered were pooled to identify the odds of the relationship between the potential independent variables and the study dependent variables. Multilevel logistics regression was used to conduct the multivariable analyses that estimated the adjusted odds ratios (AORs), which measured the strength of association with the dependent variables. Survey clusters and weights were adjusted using the STATA/MP 14.1 version ‘SVY’ command.

A stage modelling approach was adopted for the multivariable analyses, entailing that each of the seven-level factors presented in Table 1 was assessed independently. Firstly, the community-level characteristics were entered as a baseline first-stage model and those characteristics that met the 5% significance level criteria were retained (model 1). In the second-stage modelling, the significant variables in model 1 were added to the socioeconomic characteristics, and again, those variables that were significantly significant were retained in model 2. This procedure was repetitively used for the inclusion of individual maternal and child-related, knowledge of health services through (media), household decision autonomy, healthcare-related service, and immediate feeding practices in the third, fourth, fifth, sixth, and seventh stages, respectively. Variables that were statistically significant in stage model 7 are reported in the study (Appendix A). This procedure permits factors that indirectly affect child health to be satisfactorily examined without interfering with direct factors that impact child health (e.g., child’s nutritional intake and disease incidence).

## 3. Results

A weighted total of 11,313 NGZ children below 60 months old were underweight, comprising 1402 (NC), 2826 (NE), and 7085 (NW) children, over the 10-year study period. In the sub-age groups, a greater prevalence of underweight children (prevalence = 34.8%, 95% CI: 33.5–36.2) was observed among 24–59-month-old children (Figure 1a). The prevalence of underweight children in NGZ hardly decreased between 2008 and 2013; however, a decreasing trend was observed from 2013 to 2018. The decreasing underweight prevalence trend was more evident among NW children, which statistically significantly (error bar did not overlap) declined from 47.5% (44.8–50.1) in 2013 to 34.8 (32.0–37.6) in 2018 (Figure 1b).

The prevalence of underweight children who had diarrhoea episodes 14 days preceding the survey interview date slightly declined from 41.4% (38.5–44.4) in 2008 to 39.0 (35.5–42.5) in 2018. Similarly, children residing in rural areas recorded a slight decline in underweight prevalence, from 33.7% in 2008 to 31.9% in 2018. The underweight prevalence of children from poor households in NGZ remained stagnant during the study period; that is, it remained unchanged from 35.6% in 2008 to 35.5% in 2018 (Appendix A). In the NC geopolitical zone, only the Niger state recorded a steady but modest decreasing trend of underweight prevalence in both age groups (children aged between birth and 23 months (Figure 2a) and children aged from 24 to 59 months (Figure 2b)) over the 10-year study period.

The pace of underweight prevalence decrease across the six states in NE remained almost stagnant for children aged 0–23 months (Figure 3a). However, in the case of children aged 24–59 months, only two states (Adamawa and Bauchi) reported a steady modest underweight prevalence decline (Figure 3b).

A steady decreasing trend of underweight prevalence among 0–23-month-old children in the NW geopolitical zone was lacking across the seven states (Figure 4a). A similar trend was also noted for 24–59-month-old children except for Jigawa, which had a slight statistically insignificant (error bar overlapped) decrease (Figure 4b).

### 3.1. Independent Characteristics Related to Underweight among 0–23-Month-Old Children

Detailed findings of all adjusted model analyses of underweight children in NGZ are presented in Appendix A for 0–23, 24–59, and 0–59 months, respectively. As shown in Table 2, a statistically significantly higher probability of underweight children aged 0–23 months born to mothers who had no schooling (AOR = 1.63, 95% CI: 1.21–2.19) was more likely to be underweight than the children of mothers who had at least primary education. There was an increased likelihood of underweight prevalence among children from a poor (AOR = 1.53, 95% CI: 1.13–2.06) or average (AOR = 1.39, 95% CI: 1.05–1.85) household than that among children from an affluent household. Other significantly greater odds for underweight children aged 0–23 months in the NGZ were residing in NW, being a male, diarrhoea episode, child’s body size at birth as perceived by their mother, mother’s height, fever, dietary diversity score, and delivery assistance (Table 2). Collinearity assessment showed that when the delivery assistance of children aged 0–23 months was substituted by birthplace in the final model, it was observed that 0–23-month-old children delivered at non-health facility (AOR = 1.32, 95% CI: 1.01–1.72) had increased likelihood of being underweight than those delivered at a health facility.

### 3.2. Independent Characteristics Related to Underweight among 24–59-Month-Old Children

Between 24- and 59-month-old children who had diarrhoea in the 14 days preceding the survey date were more likely to be underweight (AOR = 1.80, 95% CI: 1.41–2.30) than those who did not have diarrhoea. Children born to mothers who never watched television (AOR = 1.68, 95% CI: 1.22–2.30) had increased odds of being underweight compared with those who watched television. Compared to NC, the likelihood of underweight prevalence among 24–59 months old children rose significantly by 85% for NE and 163% for NW. Likewise, children whose mothers lacked receipt of any method of contraception (AOR = 1.66, 95% CI: 1.19–2.32) had a greater probability of being underweight (Table 2). Other variables that posed an increased probability of underweight prevalence included mother’s height, second- or third-ranked children with 2-year gaps or less, and children of fourth or higher rank with a gap of 2 years or less and poor or average households. Collinearity check also indicated that when the economic status of the households of children aged 24–59 months old was replaced with the educational attainment of mothers in the final model, a significantly greater probability of being underweight was noted for children of mothers who had no schooling (AOR = 1.35, 95% CI: 1.03–1.77).

### 3.3. Independent Characteristics Related to Underweight among 0–59-Month-Old Children

As shown in Table 2, there were increased greater odds of underweight prevalence in children below 60 months old who had a fever in the 14 days before the survey interview (AOR = 1.19, 95% CI: 1.05–1.35) and those who had five or more dietary diversity intakes in the 24 h preceding the survey (AOR = 1.42, 95% CI: 1.16–1.75). Moreover, being a male child had a 1.18 times greater likelihood of underweight prevalence, and children of fourth or higher rank birth with 2-year gaps or less had 1.38 times increased odds of underweight prevalence. The multivariable results also indicated that the mother’s height, children whose body size at birth was perceived as small or smaller, children living in NE or NW, children from poor or average households, children whose mothers had no schooling, and children who had diarrhoea episodes were significantly greater likelihood of underweight.

## 4. Discussion

The estimated overall underweight prevalence of children below 60 months old in the NGZ between 2008 and 2018 was 33.5% (32.3–34.7), which is well above the most recently reported national prevalence of 21.8%. Findings from this study suggest a nutritional concern among NGZ children, particularly the sub-age group of 24–59 months, which reported the highest underweight prevalence at 34.8% (33.5–36.2). It has been indicated earlier that children below 24 months of age have a significantly lower likelihood of being underweight compared with those aged 24 months or older [23,24]. A slightly decreasing trend in the prevalence of underweight children was noted in the NGZ, especially in NC and NE, during the study period, and a similar trend was noted in the disaggregated age groups across the 19 states in the NGZ. Underweight prevalence in the NGZ remains high, and its decreasing trend is concerning; hence, retooling and formulating new zone-specific intervention initiatives are crucial.

Four variables (children living in the NE or NW, children in a poor economic or average household, having a short mother, and children who suffered diarrhoea episodes) were consistently identified as statistically significantly related to the increased likelihood of underweight prevalence across each of the study sub-age groups (0–23, 24–59, and 0–59 months). Furthermore, the higher probability of underweight prevalence among 0–23- and 0–59-month-old children was associated with those whose mothers had no formal education, being a male child, a child who had a fever 14 days before the survey date, dietary diversity intake (≥5), and small or very small body size at birth, while delivery assistance by non-health personnel was only related to 0–23-month-old children. Having a mother who never watched television and a mother who did not use any form of contraception were significantly associated with underweight prevalence in 24–59-month-old children. A fourth- or higher-rank birth with a short birth gap (2 years or less) was significantly related to 24–59- and 0–59-month-old children.

Across the three age groups, children from poor or average economic households had a greater probability of underweight prevalence compared with those residing in affluent households. This finding is in line with those of earlier studies [25,26], and a range of factors such as access to exorbitant nutritional food, residing in a decent environment with a clean source of water and improved sanitation infrastructure, access to well-equipped health facilities, and better knowledge of childcare practices could have potentially reduced the odds of underweight prevalence among children of affluent households. Location (i.e., region, province, and geopolitical zone) [18,23] has been previously shown to be significantly associated with underweight prevalence in children. Likewise, in the current study, all sub-age groups of children residing in NE or NW reported greater odds of underweight prevalence compared with those living in NC. This finding is not surprising because severe food insecurity is increasing in the two geopolitical zones. Militant banditry and insurgency and cattle rustling have disrupted a remarkable number of livestock herders and farmers from cultivating and accessing their farmlands for over 10 years, resulting in poor agricultural production. In addition, the impact of cultural preferences on food intake might have deprived children of nutritious food; instead, children are fed with native meals and traditional herbs with little or no nutrient content [27].

The likelihood of underweight prevalence among the three age groups was significantly higher for children who reported diarrhoea illness 14 days prior to the survey date than those with no diarrhoea incidence. This outcome is consistent with that of a similar study performed in Vietnam [28]. A possible clarification for this result can be linked to poor nutritional dietary intake, which often results in decreased appetite, nutrient losses due to vomiting, and impaired intestinal absorption [29]. There was a consistent significantly increased likelihood between the maternal height of 159 cm or lower and underweight prevalence among children of all age groups compared with those of mothers having a height of 160 cm or greater. This finding is similar to that obtained by Dewey et al. [30], and the consistent association can be linked to shared genetics (i.e., pelvic size and foetal programming) and common environmental factors, such as poor dietary intake and culture, which might have affected mothers in their early childhood and later their offspring [31]. This outcome suggests the need for regular weight and growth monitoring of offspring of shorter maternal height for preventive and curative healthcare.

An increased likelihood of underweight prevalence among 0–23- and 0–59-month-old children was related to mothers without formal education compared with those who attended secondary or higher education, which aligns with the previously obtained results [20,32]. This might be due to uneducated mothers being less likely to understand and use good childcare practices (e.g., timely feeding, immunization, and hygienic behaviours); in addition, they are more likely to follow cultural and religious practices that may be harmful to the child’s health. It is possible that underweight prevalence in children in NGZ might be remarkably reduced through women’s education empowerment. Male and 0–23- and 0–59-month-old children reported a significantly greater likelihood of association with underweight children compared with their female counterparts. This outcome is in contrast with that of an earlier study performed in Nepal, which suggested that being a female child was more likely to be underweight [33]. Nevertheless, the current finding is consistent with those of previous studies [18,34], and the reason for this inconsistency in gender dissimilarity in underweight children remains uncertain.

Children belonging to sub-age groups (0–23 and 0–59 months old) documented to be small or smaller based on their body size by their mothers after delivery were more predisposed to underweight prevalence compared with their large or larger counterparts. This finding aligns with those obtained previously [18,35], which indicated that small-sized children at birth were more predisposed to be underweight. Small-sized children can be linked to the mother’s inadequate nutritional intake and maternal organ size during pregnancy. Likewise, delivery assistance significantly elevated a child’s susceptibility to underweight prevalence. The 0–23-month-old children delivered by unskilled health personnel reported significantly greater odds of being underweight than those attended by skilled health personnel. A lack of adequate and appropriate postnatal counselling might have contributed to the current finding. It was also observed that 0–23- and 0–59-month-old children who adhered to five or more recommended acceptable minimum dietary diverse nutritional foods 24 h preceding the survey date were more likely to be underweight compared with those having less than five DDS. This result remains unclear; however, well-timed complementary feeding initiation and possible intake of unhealthy foods in conjunction with diverse dietary foods may have led to the surprising increased odds of adequate DDS observed. Furthermore, 0–23- and 0–59-month-old children who reported having a fever 14 days before the survey interview were more likely to be underweight than those who did not. It has been previously suggested that children who have a fever two weeks preceding the survey indicate inadequate nutritional status [36]. This is attributable to reduced appetite, resulting in aggravated undernutrition.

The sub-aged group of 24–59-month-old children living in households exposed to television had a lower likelihood of being underweight compared with those in households without access to television. Electronic or print media remain an important source of health information, as they broadcast information concerning immunisation, breastfeeding, and complementary feeding. Likewise, children of mothers who did not receive any form of contraception had a higher probability of being underweight than those whose mothers had contraception. This outcome is supported by a longitudinal study performed in Bangladesh, which indicated that the body mass index of children significantly improved after the scaled-up adoption of family-planning measures (i.e., contraceptive use) among women [37]. Rana and Goli [38] suggested that contraceptive use indirectly impacts the biological and reproductive functioning of mothers and children; for example, undernourished mothers during pregnancy have higher odds of experiencing poor birth outcomes, which often results in undernutrition among their children [39]. Moreover, 24–59- and 0–59-month-old children whose birth rank (from two to four or higher) with a short birth gap (≤2 years) were more likely to be underweight compared with those born with lengthier inter-birth spacing (>2 years). This outcome may be linked to insufficient care and negligence the higher-ranked children received as well as inadequate economic resources, particularly children residing in poor households, which often results in feeding competition among siblings—leading to malnutrition.

This study has the following limitations: First, data on dietary diversity food intake of children 24 h preceding the survey interview date might have been erroneously reported by the respondents, especially those in rural areas. Second, a causal association with the outcome variables could not be estimated because the study was based on a cross-sectional design. Third, the impact of unmeasured residual co-existing variates could have affected the current estimates (e.g., measures of child dietary intake and timely feeding pattern) because of the unavailability of data. Fourth, information concerning the medical condition of mothers and children below 60 months old was lacking during the survey interview. Fifth, underestimation or overestimation of estimates may have occurred in the study findings due to the unavailability of data concerning the status of children with long gaps in vaccination and those with incomplete vaccination, by type of vaccination. Sixth, we were unable to measure the potential impact of the monotony of diet and the minimum quality and quantity of food given to children due to the unavailability of data. Seventh, estimates obtained in the study may have been impacted due to the unavailability of household food insecurity, or household hunger score data were lacking. Eight, misinterpretation of weight for age might have a bias in the study estimates because weight gain can reflect children becoming taller, fatter, or both as previously suggested by Victora et al. [5]. The strengths of the study were that the underweight indicator used was based on the WHO’s description. Geopolitical zone-specific representative data were used to identify primary characteristics associated with underweight prevalence among 0–23-, 24–59-, and 0–59-month-old children, which will enable policymakers to effectively initiate tailored interventions to scale down underweight prevalence across NGZ. Additionally, the strength of the statistical power was very high in detecting any statistical differences because three NDHS datasets were pooled.

## 5. Conclusions

Four characteristics were established to be consistently significantly associated with underweight across the considered age group of 0–23-, 24–59-, and 0–59-month-old children in the NGZ, Nigeria. These characteristics included children living in the northeast or northwest geopolitical zone, children from poor or middle-income households, children of short mothers, and children diagnosed with diarrhoea illness 14 days before the survey. The outcomes indicate the need for individual-level interventions, and such initiatives to reduce underweight prevalence in children should focus on alleviating poverty through the transmission of cash and the well-timed monitoring of short mothers’ offspring, particularly for low socioeconomic households. Interventional initiatives at the community level should include the establishment of a clean source of drinking water and improved sanitation systems (i.e., sewage) to substantially scale down diarrhoea occurrence in areas with inadequate social structural development, such as rural communities and urban slums.

## Figures and Tables

**Figure 1 nutrients-14-02042-f001:**
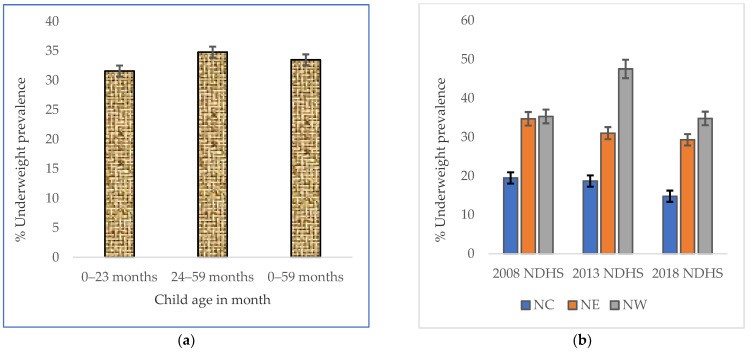
(**a**) The overall underweight prevalence in each of the sub-age groups in the NGZ, with a 95% confidence interval by age in months during the study period; (**b**) trends in prevalence of underweight children in the northern geopolitical zones (north central (NC), northeast (NE), and northwest (NW)) by year of Nigeria Demographic and Health Survey (NDHS).

**Figure 2 nutrients-14-02042-f002:**
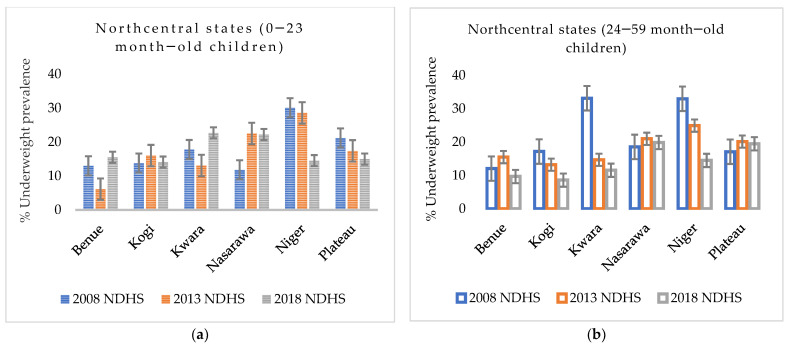
(**a**) Trends in prevalence of underweight among 0–23-month-old children from 2008 to 2018 Nigeria Demographic and Health Survey (NDHS), with a 95% confidence interval by north-central states; (**b**) trends in prevalence of underweight among 24–59-month-old children from 2008 to 2018 NDHS, with a 95% CI by northcentral states.

**Figure 3 nutrients-14-02042-f003:**
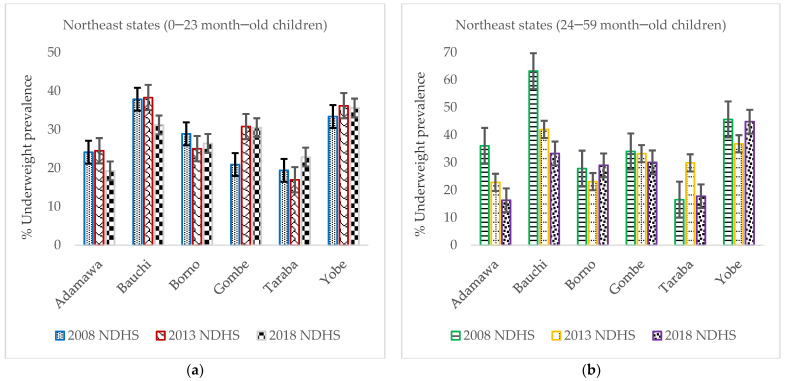
(**a**) Trends in prevalence of underweight among 0–23-month-old children from 2008 to 2018 Nigeria Demographic and Health Survey (NDHS), with a 95% confidence interval (CI) by northeast states; (**b**) trends in prevalence of underweight among 24–59-month-old children from 2008 to 2018 NDHS, with a 95% by northeast states.

**Figure 4 nutrients-14-02042-f004:**
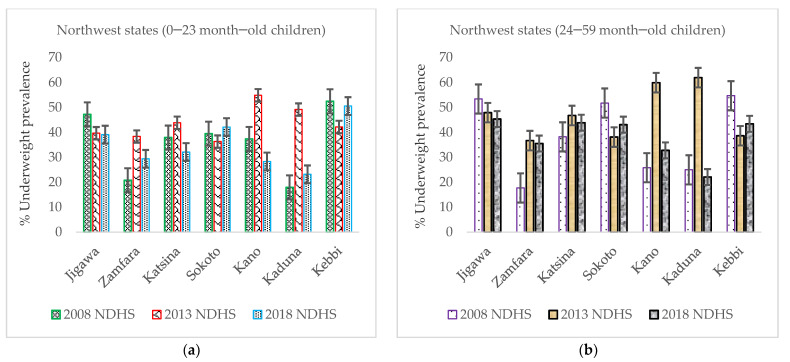
(**a**) Trends in prevalence of underweight among 0–23-month-old children from 2008 to 2018 Nigeria Demographic and Health Survey (NDHS), with a 95% confidence interval by northwest states; (**b**) trends in prevalence of underweight among 24–59-month-old children from 2008 to 2018 NDHS, with a 95% CI by northwest states.

**Table 1 nutrients-14-02042-t001:** Classification and description of likely independent confounding related characteristics to underweight in children below 60 months old for the study analysis.

Independent Characteristics	Classification	Reference Category (Rf)
Community-level		
Place of residence	1 = Urban; 2 = Rural	Urban
Location of the geopolitical zone	1 = Northcentral; 2 = Northeast; 3 = Northwest	Northcentral
Socioeconomic-level		
Household economic status	1 = Poor; 2 = Average; 3 = Rich	Rich
Educational attainment by mothers	1 = No educational attainment; 2 = Primary attainment; 3 = Secondary or higher attainment	Secondary or higher attainment
Mother’s employment status	1 = Unemployed; 2 = Employed	Unemployed
Educational attainment by fathers	1 = No educational attainment; 2 = Primary attainment; 3 = Secondary or higher attainment	Secondary or higher attainment
Household number of wives or women	1 = A wife/woman; 2 = Two or more wives/women	A wife/woman
Maternal individual level		
Age of mother at delivery (years)	1 = Less than 20; 2 = Between 20 and 29; 3 = Between 30 and 39; 4 = Between 40 and 49	Between 30 and 39 years
Maternal weight	1 = Standard (18.5 ≤ MBMI ≤ 24.9); 2 = Underweight (MBMI < 18.5); 3 = Overweight (25 ≤ MBMI ≤ 29.9); 4 = Obese (MBMI ≥ 30)	Underweight (MBMI < 18.5)
Birth control use	1 = Used contraceptive; 2 = Non-use of contraceptive	Used contraceptive
Mothers height ‡	1 = Height greater or equal to 160; 2 = Between 155 and 159; 3 = Between 150 and 154; 4 = Between 145 and 149; 5 = Less than 145	Height greater or equal to 160
Birth rank and interval of birth	1 = 2nd- or 3rd-ranked child, interval greater than 2 years; 2 = First-ranked child; 3 = 2nd- or 3rd-ranked child, interval less than or equal to 2 years; 4 = 4th- or greater-ranked child, interval more than 2 years; 5 = 4th- or greater-ranked child, interval less than or equal to 2 years	2nd- or 3rd-ranked child, interval greater than 2 years
Child individual level		
Child’s sex	1 = Female; 2 = Male	Female
Perceived baby body size at birth by mothers	1 = Middle or larger; 2 = Small or smaller	Middle or larger size
Knowledge of health services through (media)		
Occurrence of listening to a radio	1 = Once or more a week; 2 = Less than once a week; 3 = Never	Once or more a week
Occurrence of watching television	1 = Once or more a week; 2 = Less than once a week; 3= Never	Once or more a week
Occurrence of reading newspaper or magazine	1 = Once or more a week; 2 = Less than once a week; 3 = Never	Once or more a week
Household decision autonomy		
Wife has money influence	1 = Someone else or husband alone; 2= Wife alone or joint decision	Someone else or husband alone
Wife has healthcare influence	1 = Someone else or husband alone; 2 = Wife alone or joint decision	Someone else or husband alone
Wife has movement influence	1 = Someone else or husband alone; 2 = Wife alone or joint decision	Someone else or husband alone
Healthcare-related service		
Delivery place	1 = Home; 2 = Healthcare institution	Healthcare institution
Type of delivery	1 = Vaginal delivery; 2 = Caesarean delivery	Vaginal delivery
Birth assistance	1 = Skilled health personnel; 2 = Non-health personnel	Skilled health personnel
Immediate feeding practices		
Dietary diversity score (DDS)	1 = DDS less than 5 foods/inadequate; 2= DDS Greater or equal to foods/adequate	DDS less than 5 foods/inadequate
Breastfeeding initiation	1 = Within one hour of delivery; 2 = Greater than one hour after delivery	Within one hour of delivery
Presently breastfeeding	1 = Not breastfeeding; 2 = Breastfeeding now	Not breastfeeding
Length of breastfeeding	1 = Up to 12 months; 2 = Greater than 12 months	Up to 12 months
Full vaccination	Yes, a child received vaccination; No otherwise	No
The child had diarrhoea in the last 14 days before the survey interview	Yes, if a child had diarrhoea; No otherwise	No
The child had a fever in the last 14 days before the survey interview	Yes, if a child had a fever; No otherwise	No

Notes: ‡, maternal height measured in centimetres; MBMI, maternal body mass index (estimated in kilograms/square meters).

**Table 2 nutrients-14-02042-t002:** Adjusted odds ratios for characteristics significantly related to underweight among 0–23 m, 24–59 m, and 0–59 m old children in the NGZ, Nigeria.

Characteristics	Underweight Child ¥ 0–23 m	Underweight Child ¥ 24–59 m	Underweight Child ¥ 0–59 m
Community-level			
Place of residence			
Urban	-		-
Rural	-	-	-
Location of the geopolitical zone			
Northcentral	Rf	Rf	Rf
Northeast	1.33 (0.98–1.81)	1.85 (1.40–2.44)	1.59 (1.30–1.95)
Northwest	1.64 (1.20–2.23)	2.63 (2.03–3.42)	2.19 (1.78–2.68)
Socioeconomic level			
Household economic status			
Rich	Rf	Rf	Rf
Average	1.39 (1.05–1.85)	1.57 (1.19–2.07)	1.58 (1.27–1.95)
Poor	1.53 (1.13–2.06)	1.64 (1.22–2.20)	1.67 (1.33–2.11)
Educational attainment by mothers			
Secondary or higher attainment	Rf	-	Rf
Primary attainment	1.25 (0.86–1.79)	-	1.10 (0.87–1.40)
No educational attainment	1.63 (1.21–2.19)	-	1.55 (1.28–1.87)
Mother’s employment status			
Unemployed	-	-	-
Employed	-	-	-
Educational attainment by fathers			
Secondary or higher attainment	-	-	-
Primary attainment	-	-	-
No educational attainment	-	-	-
Household number of wives or women			
A wife/woman	-	-	-
Two or more wives/women	-	-	-
Maternal individual level			
Age of mother’s delivery (years)			
Less than 20	-	-	-
Between 20 and 29	-	-	-
Between 30 and 39	-	-	-
Between 40 and 49	-	-	-
Maternal weight			
Underweight (MBMI < 18.5)	-	Rf	Rf
Standard (18.5 ≤ MBMI ≤ 24.9)	-	0.56 (0.44–0.71)	0.57 (0.48–0.68)
Overweight or Obese (25 ≤ MBMI ≤ 29.9)/(MBMI ≥ 30)	-	0.41 (0.29–0.57)	0.41 (0.32–0.53)
Birth rank/interval of birth			
First-ranked child	-	0.95 (0.71–1.27)	1.04 (0.84–1.28)
2nd- or 3rd-ranked child, interval greater than 2 years	-	1.43 (1.03–1.99)	1.27 (0.97–1.67)
2nd- or 3rd-ranked child; interval less than or equal to 2 years	-	Rf	Rf
4th- or greater-ranked child, interval more than 2 years	-	1.17 (0.91–1.49)	1.18 (0.99–1.40)
4th- or greater-ranked child, interval less than or equal to 2 years	-	1.46 (1.08–1.98)	1.38 (1.10–1.73)
Birth control use			
Use of contraceptive	-	Rf	-
Non-use of contraceptive	-	1.66 (1.19–2.32)	-
Maternal height (centimetre (CM))			
Height greater or equal to 160	Rf	Rf	Rf
Between 155 and 159	1.20 (0.93–1.54)	1.41 (1.14–1.75)	1.35 (1.14–1.59)
Between 150 and 154	1.38 (1.06–1.79)	1.72 (1.36–2.17)	1.60 (1.35–1.89)
Between 145 and 149	1.55 (1.06–2.27)	2.22 (1.58–3.12)	1.92 (1.48–2.49)
Less than 145	3.49 (1.69–7.23)	1.81 (1.38–3.99)	2.18 (1.24–3.82)
Child individual level			
Child’s sex			
Female	Rf	-	Rf
Male	1.50 (1.24–1.82)	-	1.18 (1.04–1.34)
Perceived baby body size at birth by their mothers			
Middle or larger	Rf	-	Rf
Small or smaller	1.85 (1.45–2.36)	-	1.54 (1.30–1.84)
Knowledge of healthcare services through media			
Occurrence of listening to a radio			
Once or more a week	-	-	-
Less than once a week	-	-	-
Never	-	-	-
Occurrence of reading newspaper or magazine			
Once or more a week	-	-	-
Less than once a week	-	-	-
Never	-	-	-
Occurrence of watching television			
Once or more a week	-	Rf	-
Less than once a week	-	1.10 (0.76–1.58)	-
Never	-	1.68 (1.22–2.30)	-
Household decision autonomy			
The wife has earning influence			
Someone else or partner/husband	-	-	-
Wife alone or joint decision	-	-	-
The wife has a healthcare influence			
Someone else or partner/husband	-	-	Rf
Wife alone or joint decision	-	-	0.78 (0.66–0.92)
influence			
Someone else or partner/husband	-	-	-
Wife alone or joint decision	-	-	
Healthcare-related services			
Delivery place			
Healthcare institution	-	-	-
Home	-	-	-
Type of delivery			
Vaginal delivery			-
Caesarean	-	-	
Birth attendant			
Skilled health personnel	Rf	-	-
Unskilled personnel	1.46 (1.13–1.89)	-	
Immediate feeding practices			-
Dietary diversity score (DDS) ‡			
DDS less than 5 foods/inadequate	Rf	-	Rf
DDS greater or equal to 5 foods/adequate	1.48 (1.20–1.81)	-	1.42 (1.16–1.75)
Breastfeeding initiation ‡			
Greater than 1 h after delivery	-	-	-
Within 1 h of delivery	-	-	-
Presently breastfeeding ‡			
Not breastfeeding	-	-	Rf
Breastfeeding now	-	-	0.78 (0.66–0.91)
Length of breastfeeding ‡			
Up to 12 months	Rf	-	-
More than 12 months	0.47 (0.24–0.93)	-	-
Full vaccination			
No	-	-	-
Yes	-	-	-
The child had diarrhoea in the last 14 days before the survey interview			
No	Rf	Rf	Rf
Yes	1.42 (1.13–1.78)	1.80 (1.41–2.30)	1.59 (1.35–1.87)
The child had a fever in the last 14 days before the survey interview			
No	Rf	-	Rf
Yes	1.26 (1.03–1.55)	-	1.19 (1.05–1.35)

Notes: m, months; Rf, reference group; NGZ, combined geopolitical zones (northcentral, northeast, and northwest); MBMI, maternal body mass index (estimated in kilograms per square meter); ¥, adjusted odds ratios with a 95% corresponding confidence interval for independent characteristics; ‡, independent characteristics that were not adjusted for children aged 24–59 months.

## Data Availability

This study was based on a public domain dataset that is freely available online: https://dhsprogram.com/data/dataset/Nigeria_Standard-DHS_2018.cfm?flag=0; https://dhsprogram.com/data/dataset/Nigeria_Standard-DHS_2013.cfm?flag=1 and https://dhsprogram.com/data/dataset/Nigeria_Standard-DHS_2008.cfm?flag=1 (accessed on 15 October 2021).

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
