# Peer review of "Factors Related to Underweight Prevalence among 33,776 Children Below 60 Months Old Living in Northern Geopolitical Zones, Nigeria (2008–2018)"

_nutrients, 2022, doi:10.3390/nu14102042_

Round 1

Reviewer 1 Report

This study investigated the likely influencing factor on underweight children below 60 months, considering geopolitical zone-effect and disaggregated in different age groups, based on Nigeria DHS in NGZ in the year 2008, 2013 and 2018. The logic of the study is clear and the conceptual framework for potential influencing factors is reasonable. The outcomes make evidence-based support on interventions of children underweight in the study region. However, there are several points needed to clarify as following:

(1) The title stated “disaggregated analysis at geopolitical zone-level”. However, the geopolitical zone-level was put as a fixed effect in the model, without any considering interactions with other potential influencing factors, or without separate analysis according to different geopolitical zones. The title should be revised according to the actually methodologies and results.

(2) Data analysis was based on data from NGZ, thus the title and the main text should focus on NGZ instead of the whole Nigeria.

(3) Abstract should mention data source of this study.

(4) The authors separated children into two age groups: 0-23 months and 24-59 months. Why not 0-36 months and 27-59 months old? Please clarify the reasons to make this cut-off.

(5) The authors analyzed data separately on the two different age groups as well as on children with aggregated ages. In fact, if the authors assumed that the two separated age groups have different influencing factors, which was also shown in results, what are the reasons to combine the two groups, without adding age and age-factor interaction terms in the model?

(6) The authors transformed continuous covariates into categorical ones in the model. Why didn’t them consider the original continuous from of these variables? The transformation may lead to loss of information. Please clarify it.

(7) The authors stated that they used “multilevel logistic regression”. However, “multilevel” models were used to analyze data with hierarchical structure. For example, if individuals within the same households or the same clusters were not independent with each other, household- or cluster-level random effects should be put into the models for accounting the dependence of individuals. However, the authors assumed individuals independent and all covariates were put as fixed effect factors in the models. Such models were not “multilevel regressions”.

(8) The authors used a stage modelling approach for selection of covariates in the final model. What is the reason to begin with community level factors, and then socioeconomic factors, and then individual level factors, and so on? Different inclusion orders may result in different covariates selected in the final model. In addition, did the authors test the correlations between covariates to avoid collinearity, even though covariates were all transformed to categorical ones?

(9) Why wasn’t the temporal effect taken into account in the final models? Different geopolitical zones show different temporal trends of underweight prevalence, thus, zone-time interaction may exist.

(10) There were several factors differently associated with the risk of underweight in different separated age groups, for example, educational attainment by mothers were found significant in children 0-23 months, but not in children 24-59 months, while birth rank/interval of birth were found significant in children 24-59 months but not in that of 0-23 months. What are the possible reasons for these differences and how can they imply for interventions on underweight in different age groups?

(11) One major focus of the manuscript was the risks of underweight in disaggregated geopolitical zones. However, how risk differences in geopolitical zones imply on interventions were not fully discussed.

Author Response

Comment

Response

(1) The title stated “disaggregated analysis at geopolitical zone-level”. However, the geopolitical zone-level was put as a fixed effect in the model, without any considering interactions with other potential influencing factors, or without separate analysis according to different geopolitical zones. The title should be revised according to the actually methodologies and results.

Thanks,

for the clarity it now reads;

Title:

“Factors related to underweight prevalence among 33,776 children below 60 months old living in northern geopolitical zones, Nigeria (2008-2018)”.

(2) Data analysis was based on data from NGZ, thus the title and the main text should focus on NGZ instead of the whole Nigeria.

Thanks, the title was updated as stated above. Also, the last paragraph of the manuscript introduction was updated to reflect reviewer’s suggestion.

(3) Abstract should mention data source of this study

Source of data (Nigeria demographic and health survey (NDHS) now included in the abstract section

(4) The authors separated children into two age groups: 0-23 months and 24-59 months. Why not 0-36 months and 27-59 months old? Please clarify the reasons to make this cut-off.

We did not consider 0-36 months and 27-59 months old as suggested by the reviewer because undernutrition or poor fetal growth in the first 2 years of life leads to irreversible damage [1]. Hence interventions to reduce undernutrition or poor fetal growth could either start from 0-23 months or 0-59 months.

It is also important to note that 0-23 months of interventions are primarily breastfeeding and complementary feeding practices. While that of 0-59 months interventions include breastfeeding, complementary feeding practices plus Psychosocial stimulation [2].

References:

1). Victora, C.G., Adair, L., Fall, C., Hallal, P.C., Martorell, R., Richter, L., Sachdev, H.S. and Maternal and Child Undernutrition Study Group, 2008. Maternal and child undernutrition: consequences for adult health and human capital. The lancet, 371(9609), pp.340-357.

2). Hamadani, J.D., Huda, S.N., Khatun, F. and Grantham-McGregor, S.M., 2006. Psychosocial stimulation improves the development of undernourished children in rural Bangladesh. The Journal of nutrition, 136(10), pp.2645-2652.

(5) The authors analyzed data separately on the two different age groups as well as on children with aggregated ages. In fact, if the authors assumed that the two separated age groups have different influencing factors, which was also shown in results, what are the reasons to combine the two groups, without adding age and age-factor interaction terms in the model?

We considered two different age groups because interventions to reduce undernutrition or poor fetal growth could either start from 0-23 months or 0-59 months and the reasons for this have been stated above.

(6) The authors transformed continuous covariates into categorical ones in the model. Why didn’t them consider the original continuous from of these variables? The transformation may lead to loss of information. Please clarify it.

Yes, we did.

The simple reason is that we followed the NDHS classification of variables in their survey dataset. In a limited resource setting like Nigeria, the classification can provide areas for immediate intervention initiatives that will earn more marginal benefits to the communities.

(7) The authors stated that they used “multilevel logistic regression”. However, “multilevel” models were used to analyze data with hierarchical structure. For example, if individuals within the same households or the same clusters were not independent with each other, household- or cluster-level random effects should be put into the models for accounting the dependence of individuals. However, the authors assumed individuals independent and all covariates were put as fixed effect factors in the models. Such models were not “multilevel regressions”.

A well-established NDHS survey design that employed a multistage, stratified, cluster random sampling method to gather data actually aimed to minimize selection bias error. Weight factors were incorporated to account for the complex clustered survey design.

Selection bias is highly unlikely to skew the findings given the random selection of participants and correction for sampling within communities and households. The STATA ‘SVY’ command was used to adjust for the complex sampling design and possible clustering of participants within each sample area.

We think fixed effect modelling is appropriate for this study and our reasons are below.

(1) Fixed effect modelling is more appropriate for this type of modelling because it involved a higher-level unit and particularly due to the large sample size within the country [3]

(2) Fixed effects models eliminate any variation in higher-level units in the estimated coefficients [4]

(3) Fixed effect estimates a causal relationship [5]

4) As indicated in points 1 and 2 above if researchers have concerned that unobserved higher-level variables may affect the estimation of this relationship, the fixed-effect model is more appropriate than the random effect [5]

References:

3) Mohring, K (2012) The fixed effect as an alternative to multilevel analysis for cross-national analyses, GK Soclife working paper.

4) Allison, P. (2009) Fixed Effects Regression Models, Sage Quantitative Applications in the Social Sciences, vol. 160.

5) Clarke, P., Crawford, C., Steele, F. & Vignoles, A. (2010) The choice between fixed and random effects models: some considerations for educational research, Institute of Education DoQSS Working Paper No. 10-10

(8) The authors used a stage modelling approach for selection of covariates in the final model. What is the reason to begin with community level factors, and then socioeconomic factors, and then individual level factors, and so on? Different inclusion orders may result in different covariates selected in the final model. In addition, did the authors test the correlations between covariates to avoid collinearity, even though covariates were all transformed to categorical ones?

This approach allows distal factors to be adequately investigated without meddling from proximal factors (e.g., a child’s nutrition and disease occurrence are referred to as direct or immediate factors).

We tested for collinearity in the final model, particularly place of residence-wealth, mother educational attainment- wealth, & birthplace-delivery assistance.

For clarity, the texts below are included in the result section:

Collinearity assessment showed that when the delivery attendant of children aged 0-23 months was substituted by birthplace in the final model, it was observed that 0-23 months old children delivered at non-health facility (aOR = 1.32, 95% CI: 1.01–1.72) had increased likelihood of being underweight than those delivered at a health facility.

Collinearity check also indicated that when the economic status of the households of children aged 24-59 months old was replaced with the educational attainment of mothers in the final model, a significantly greater probability of being underweight was noted for children of mothers who had no schooling (aOR = 1.35, 95% CI: 1.03–1.77).

(9) Why wasn’t the temporal effect taken into account in the final models? Different geopolitical zones show different temporal trends of underweight prevalence, thus, zone-time interaction may exist.

Our understanding is that temporal is related to time and the geopolitical zones are in the same time zone. Hence there is no need for zone-time interaction.

(10) There were several factors differently associated with the risk of underweight in different separated age groups, for example, educational attainment by mothers were found significant in children 0-23 months, but not in children 24-59 months, while birth rank/interval of birth were found significant in children 24-59 months but not in that of 0-23 months. What are the possible reasons for these differences and how can they imply for interventions on underweight in different age groups?

Uneducated mothers are more likely to strictly adhere to socio-cultural practices that may negatively affect the baby’s health (e.g., cultural prelacteal feeding practices that deprive newborns of colostrum or the first breast fluid that is rich in nutrients and immunoglobulins- thus increasing the risk of infant morbidity, particularly in the first 6 months [6,7]. As with earlier studies, the type of prelacteal feeds given to infants is related to the culture and belief system of the nursing mothers [8,9]. This may have increased the odds of being underweight among children aged 0-23 months observed.

While in the case of the significant odds observed for birth rank/interval, in children 24-59 months but not in that of 0-23 months, this could be attributed to the shorter gaps between births and pregnancies, which often lead to inattention and improper care for higher-order children’s needs. This can adversely affect a child’s health (e.g., wasting, stunting, and being underweight). Previous studies have suggested that at least three years longer child spacing is more likely to protect against child malnutrition [10,11].

Designing effective interventions in different age groups, particularly in a limited resource setting like Nigeria is very crucial. This will offer policymakers an opportunity to provide an adequate allocation of resources to the age group that may have a wider population reduction in underweight on the total number of underweight children among the NGZ population. Additionally, policymakers can target all the characteristics that were consistently related to the age groups based on the availability of resources.

References

6). Bekele, Y.; Mengistie, B.; Mesfine, F. Prelacteal feeding practice and associated factors among mothers attending immunization clinic in Harari region public health facilities, Eastern Ethiopia. Open J. Prev. Med. 2014, 4, 529–534.

7) Nguyen, P.H.; Keithly, S.C.; Nguyen, N.T.; Nguyen, T.T.; Tran, L.M.; Hajeebhoy, N. Prelacteal feeding practices in Vietnam: Challenges and associated factors. BMC Public Health 2013.

8) Khanal, V.; Adhikari, M.; Sauer, K.; Zhao, Y. Factors associated with the introduction of prelacteal feeds in Nepal: Findings from the Nepal demographic and health survey 2011. Int. Breastfeed. J. 2013.

9) Hossain, M.M.; Radwan, M.M.; Arafa, S.A.; Habib, M.; DuPont, H.L. Pre-lacteal infant feeding practices in rural Egypt. J. Trop. Pediatr. 1992, 38, 317–322.

10) Rutstein, S.O. Effects of preceding birth intervals on neonatal, infant and under-five years mortality andnutritional status in developing countries: Evidence from the demographic and health surveys. Int. J. Gynecol. Obstet. 2005, 89, S7–S24. [CrossRef]

11) Olinto, M.T.; Victora, C.G.; Barros, F.C.; Tomasi, E. Determinants of malnutrition in a low-income population: Hierarchical analytical model. Cad. Saude Publica 2004, 9, S14–S27

(11) One major focus of the manuscript was the risks of underweight in disaggregated geopolitical zones. However, how risk differences in geopolitical zones imply on interventions were not fully discussed.

It is worthy to note that we did not examine the risk of being underweight in each of the three zones because there were few recorded numbers of underweight children, and a smaller proportion may produce an estimate with a wider confidence interval.  However, we paired the three northern geopolitical zones (NC, NE, and NW) for more statistical power in assessing the risk of being underweight among children under-five years old because NGZ shares similar characteristics (i.e., socio-economic, ethnic, cultural and religious beliefs).  The NGZ population is predominantly one ethnic group (>80%). This will assist in developing workable and effective tailored evidence-based interventions across the three zones. The finding was discussed in paragraph 3 of the discussion section.

In the disaggregated geopolitical zones and their states, trends in the prevalence of being underweight among children were analyzed reported, and highlighted in the first paragraph of the discussion section

Reviewer 2 Report

Dear Authors, 

Thank you very much for preparing this very interesting manuscript. 

Please find below some comments.

Introduction

Lines 47-49. Can you please check the numbers there? You say there was a reduction of 8.3 % however your other numbers are 24% and 22%. So is it a 2% reduction? 

Methods/Results

Was the vaccination status available for those children? Vaccination status is a very important variable affecting the presence of fever and diarrhoeal episodes. 

Discussion

You refer to the results in terms of areas but you are not really discussing these results. Why is this the case that children born in certain areas are more prone to develop malnutrition? 

Reviewer 3 Report

Thank you for the opportunity to review this manuscript. Although relevant to the Nigerian context, this topic seems redundant and superfluous to the extant literature on underweight. Additionally, it is missing some important conceptual frameworks and understanding of the biology of underweight, time-trends and changing status of underweight as a marker of population nutrition status and related covariates, which must be accounted for. I have had an opportunity to read the manuscript, although the comments I wish to make are general and can be incorporated appropriately as per the authors' review:

  • Need to list cross-sectional nature of studies as a limitation, may be better to follow groups of children over time to track incidence, but given it is national surveys, you may at some point wish to comment on the transient nature of underweight as a nutrition indicator.
  • Out of curiosity, how does one account for the seasonal nature of underweight as a nutrition indicator and changes in prevalence and incidence of undernutrition based on seasonal and climatic variations? This hasn't been described and there is no real discussion of how this may also play in as a risk factor for, confounder of the associations and/or the need to account for seasonality in terms of dietary diversity and intake? 
  • Important to think about not only in terms of food availability but also in terms of monotony of diet, dietary diversity and minimum diet quality, quantity = all would have an impact on underweight.

May also want to look at household hunger scales to look at household food insecurity: https://www.fantaproject.org/monitoring-and-evaluation/household-hunger-scale-hhs

Same as above - household hunger score if you are able to extract primary variables from available data

  • Dietary diversity score (not diversity diet score) 
  • For modelling approach, please define as "forward inclusion", currently unclear how covariates were added or selected. 
  • Section 3.2: This comparison is a little meaningless and unclear why you would make it - in terms of proximal and distal determinants of underweight, they have no association with each other. It is expected that during illness episodes, child feeding may be impacted which can lead to a loss of weight. Maternal TV watching as a proxy for time-allocation and/or further downstream as a proxy for SES is not a worthwhile comparison to draw. 
  • There needs to be some significant reading of the nutrition literature - particularly the science of underweight and undernutrition. Please look at the Lancet Series 2021, 2013 and 2008 for more. Additionally, please consider reading work from the Ki cohorts, Andrew Prentice and Zulfiqar Bhutta/Cesar Victora and others who have published prolifically in the scientific literature about this topic, to help situate your analysis in the context of Sub-Saharan Africa and other relevant LMICs 
  • Additionally, there is not enough reference to the importance of evaluating underweight in the context of the first 1000 days. 
  • https://www.nestlenutrition-institute.org/nniw-95---building-future-health-and-well-being-of-thriving-toddlers-and-young-children/growth-faltering-underweight-and-stunting
